# Research Progress toward Room Temperature Sodium Sulfur Batteries: A Review

**DOI:** 10.3390/molecules26061535

**Published:** 2021-03-11

**Authors:** Yanjie Wang, Yingjie Zhang, Hongyu Cheng, Zhicong Ni, Ying Wang, Guanghui Xia, Xue Li, Xiaoyuan Zeng

**Affiliations:** 1National and Local Joint Engineering Laboratory for Lithium-Ion Batteries and Materials Preparation Technology, Key Laboratory of Advanced Battery Materials of Yunnan Province, Faculty of Metallurgical and Energy Engineering, Kunming University of Science and Technology, Kunming 650093, China; 18298325286@163.com (Y.W.); zhangyingjie09@126.com (Y.Z.); ni_zhicong@163.com (Z.N.); pzhuxsc123@163.com (Y.W.); xiagh1994@163.com (G.X.); 2National and Local Joint Engineering Laboratory for Lithium-Ion Batteries and Materials Preparation Technology, Key Laboratory of Advanced Battery Materials of Yunnan Province, Faculty of Materials Science and Engineering, Kunming University of Science and Technology, Kunming 650093, China; 17865161886@163.com; 3College of Intelligent Manufacture, PanZhihua University, Panzhihua 617000, China

**Keywords:** room temperature sodium-sulfur battery, shuttle effect, electrode design, separator, electrolyte

## Abstract

Lithium metal batteries have achieved large-scale application, but still have limitations such as poor safety performance and high cost, and limited lithium resources limit the production of lithium batteries. The construction of these devices is also hampered by limited lithium supplies. Therefore, it is particularly important to find alternative metals for lithium replacement. Sodium has the properties of rich in content, low cost and ability to provide high voltage, which makes it an ideal substitute for lithium. Sulfur-based materials have attributes of high energy density, high theoretical specific capacity and are easily oxidized. They may be used as cathodes matched with sodium anodes to form a sodium-sulfur battery. Traditional sodium-sulfur batteries are used at a temperature of about 300 °C. In order to solve problems associated with flammability, explosiveness and energy loss caused by high-temperature use conditions, most research is now focused on the development of room temperature sodium-sulfur batteries. Regardless of safety performance or energy storage performance, room temperature sodium-sulfur batteries have great potential as next-generation secondary batteries. This article summarizes the working principle and existing problems for room temperature sodium-sulfur battery, and summarizes the methods necessary to solve key scientific problems to improve the comprehensive energy storage performance of sodium-sulfur battery from four aspects: cathode, anode, electrolyte and separator.

## 1. Introduction

With the development of society and the depletion of natural resources, people have to start using renewable energy to develop low-cost and high-efficiency energy storage devices, such as secondary batteries. The ideal performance characteristics of energy storage devices are high energy density, high power density, long cycle life, low cost and high safety [1]. Among the existing secondary batteries, lithium-ion batteries have been industrialized, but their high cost, low practical energy density (100–200 Wh kg^−1^) and poor safety performance limit their application [2,3]. In order to meet the energy storage needs of current society, it is necessary to design and develop other batteries with lower cost, longer cycle life and higher energy density and power density.

Sodium is a low-cost alternative to lithium. The content of sodium in the Earth’s crust and water is 28,400 mg kg^−1^ and 1000 mg L^−1^, respectively, which far exceeds the content of lithium. The electrochemical reduction potential of sodium is −2.71 V, which is slightly higher than that of lithium (−3.02 V) [4], and is similar to the standard hydrogen electrode (SHE) potential [5]. When sodium is used as the anode, it can provide a battery voltage greater than 2 V when matched with an appropriate cathode. The high content, low cost and ability to provide high voltage make sodium an ideal choice for the anode materials of high-energy secondary batteries [6]. Sulfur has the advantages of strong oxidizing property, mature treatment technology, low cost, ready use [7], no toxicity and high capacity (when each atom transfers two electrons [8], the capacity of sulfur is 1.675 mAh g^−1^) [9], etc. Sulfur has an attractive advantage over lithium as a battery cathode. Compared with lithium-sulfur batteries, sodium-sulfur batteries are a better choice from the perspective of sustainable development and economy, or from the perspective of battery preset performance [10].

The earliest sodium-sulfur battery was constructed in the laboratory of Ford Motor Company, and Kummer and Weber confirmed its feasibility [11]. The battery uses sodium and sulfur as the active materials for the cathodes and anodes, and β-Al_2_O_3_ ceramics are used as both the electrolyte and the separator. In order to reduce the transmission resistance of sodium ions in the alumina solid electrolyte, it is necessary to ensure that the electrode material is in a molten state, so the working temperature is set at 250–300 °C. Due to the advantages of long service life, high charging efficiency and high energy density, high-temperature sodium-sulfur battery systems have been used in stationary energy storage systems [12]. However, in order to maintain the molten conductive state of the two poles, a high operating temperature is required. The high operating temperature not only causes a loss of electrical energy, but also may cause the failure of the solid electrolyte, which causes explosions and fires due to contact between the cathode and the anode. These problems limit the wide application of high-temperature sodium–sulfur batteries [13].

In order to obviate the above problems, research has been directed toward the development of room temperature sodium-sulfur batteries. The first room temperature sodium-sulfur battery developed showed a high initial discharge capacity of 489 mAh g^−1^ and two voltage platforms of 2.28 V and 1.28 V [14]. The sodium-sulfur battery has a theoretical specific energy of 954 Wh kg^−1^ at room temperature, which is much higher than that of a high-temperature sodium–sulfur battery. Although room temperature sodium-sulfur batteries solve the problems of explosion, energy consumption and corrosion of high-temperature sodium-sulfur batteries, their cycle life is much shorter than that associated with high-temperature sodium-sulfur batteries. For a wider range of applications, its cycle performance needs to be improved [13].

Room temperature sodium-sulfur batteries have the advantages of high safety performance, low cost, abundant resource and high energy density [15,16]. They not only solve the safety problem of high-temperature sodium-sulfur batteries, but also solve the problem of high cost of lithium-ion batteries, and have received widespread attention. Like the lithium-sulfur battery system, room temperature sodium-sulfur batteries also face many problems, such as:

(1) Low conductivity of sulfur (5 × 10^−30^ S·cm^−1^) and significant volume expansion (180%) [17]; (2) capacity attenuation caused by the dissolution of intermediate polysulfide in the electrolyte; (3) short circuit caused by sodium dendrites piercing the separator; (4) low utilization rate of the cathode; (5) poor reversibility, etc. [1]. This article will start with a description of the electrochemical reaction mechanism for the room temperature sodium-sulfur battery, and describe the development of room temperature sodium-sulfur battery in recent years in terms of its cathode, electrolyte, separator design and anode protection.

## 2. Electrochemical Reaction Mechanism

The sodium-sulfur battery realizes the conversion between chemical energy and electrical energy through the electrochemical reaction between metallic sodium and elemental sulfur [18]. When discharging, sodium metal produces Na^+^ and electrons. Na^+^ moves with the electrolyte through the separator to the sulfur cathode. Elemental sulfur is reduced and combined with sodium to form sodium polysulfide [19]. When a certain voltage is applied to the external circuit, the reverse reaction of the decomposition of sodium polysulfide into metallic sodium and sulfur will occur. Figure 1 is a typical room temperature sodium-sulfur battery charge/discharge curve, with two potential platforms of 2.20 V and 1.65 V during discharge, and two potential slope discharge regions within the potential range of 2.20–1.65 V and 1.60–1.20 V. There are two potential platforms of 1.75 V and 2.40 V when charging. The above redox process corresponds to the cyclic voltammetry curve of a sodium-sulfur battery. In Figure 1, the two reduction peaks at 2.20 V and 1.65 V correspond to two discharge platforms, and the two oxidation peaks at 1.75 V and 2.40 V correspond to two charging platforms [8].

Among the more than 30 solid allotropes of elemental sulfur, the ring-shaped crown-shaped sulfur eight molecule (S_8_) is the most common and stable [20]. According to the above redox process, the total reaction of the sodium-sulfur battery is:(1)$$2\mathrm{Na}\;+\;\frac{n}{8}S_{8}\;↔\;{\mathrm{Na}}_{2}S_{n}\;(1\;\leq \;n\;\leq \;8)$$

The actual charging and discharging process of room temperature sodium–sulfur battery is far more complicated than the above reaction (Figure 2). First, the battery reaction involves a multistep reaction, which will produce a variety of polysulfide intermediate products with different chain lengths. Second, the shuttle effect makes the system more complex, and the initial formation of soluble long-chain polysulfide (Na_2_S_x_ (4 ≤ x ≤ 8)), as the electrolyte diffuses to the anode, it is reduced to produce insoluble short-chain polysulfides (Na_2_S_x_ (1 ≤ x ≤ 3)) [2].

The short-chain polysulfide on the cathode side will also diffuse to the anode due to the effect of the electric field and the concentration difference and be reoxidized to long-chain polysulfide. The polysulfide dissolved in the electrolyte moves back and forth between the cathodes and anodes with the electrolyte, which is the shuttle effect [21]. It not only consumes active materials and reacts with metallic sodium, but also generates insoluble short-chain polysulfides that are deposited on the surface of the anode, hindering the transmission of electrons, resulting in low coulombic efficiency and reversible capacity of room temperature sodium-sulfur batteries.

By analyzing the X-ray photoelectron spectroscopy (Figure 3a,b) of the cathode after discharge, the final discharge product of the sodium-sulfur battery can be determined. The results show that after the battery was discharged, most of the sulfur was reduced. Combined with UV-visible light absorption spectroscopy (Figure 3c), it has proved the rationality that various forms of polysulfides can coexist in equilibrium [22]. Further through the calculation of the enthalpy of formation (DH), the thermodynamic stability of Na_2_S_5_, Na_2_S_4_, Na_2_S_2_ and Na_2_S and the metastability of Na_2_S_3_ were verified. Additionally, Na_2_S_3_ can be decomposed into Na_2_S_2_ and Na_2_S_4_ [23]. Theoretical research on S, Na_2_S_5_, Na_2_S_4_, Na_2_S_2_ and Na_2_S through first-principles shows that these Na-S crystals are all potential products of room temperature sodium-sulfur battery charging and discharging. The voltage curve of the Na concentration on the cathode of the sodium-sulfur battery (Figure 4a) was calculated by PBE-D2 (assuming that the Na concentration is the dependent variable of the discharge reaction), and the calculated main voltage regions were 2.09–2.11 V (Na_2_S_5_ and Na_2_S_4_), 1.79 V (Na_2_S_2_) and 1.68 V (Na_2_S) corresponding to the formation of the above-mentioned polysulfides [24,25,26]. Figure 4b shows the discharge curve of a room temperature sodium-sulfur battery [8]. It can be seen that S_8_ has undergone a series of complex changes, completing the transformation from solid-phase simple substance to liquid-phase long-chain polysulfide and then to insoluble short-chain polysulfide. From the analysis of thermodynamics and phase transition (the solid vertical line in the figure represents the theoretical capacity of each product), the discharge process can be divided into four parts according to the reaction steps.
S_8_ (S) + 2Na^+^ + 2e^−^ → Na_2_S_8_ (L)(2)

The solid-liquid transition from S_8_ to Na_2_S_8_ corresponds to the high voltage region of 2.20 V, and elemental sulfur is reduced to molten Na_2_S_8_.
Na_2_S_8_ (L) + 2Na^+^ + 2e^−^ → 2Na_2_S_4_ (L) (reactions that may be involved during the period: Na_2_S_8_ (L) + 2/3Na^+^ + 2/3e^−^ → 4/3Na_2_S_6_ (L); Na_2_S_6_ (L) + 2/5Na^+^ + 2/5e^−^ → 6/5Na_2_S_5_ (L); Na_2_S_5_ (L) + 1/2Na^+^ + 1/2e^−^ → 5/4Na_2_S_4_ (L))(3)

Na_2_S_8_ is reduced to Na_2_S_x_ (4 ≤ x ≤ 5), and the most important product is Na_2_S_4_, which corresponds to the inclined area of 2.20–1.65 V. This area is most complicated by the chemical balance between various polysulfides in the solution.
Na_2_S_4_ (L) + 2/3Na^+^ + 2/3e^−^ → 4/3Na_2_S_3_ (S); Na_2_S_4_ (L) + 2Na^+^ + 2e^−^ → 2Na_2_S_2_ (S); Na_2_S_4_ (L) + 6Na^+^ + 6e^−^ → 4Na_2_S (S)(4)

Na_2_S_3_, Na_2_S_2_ and Na_2_S are generated from Na_2_S_4_, corresponding to a low voltage plateau of 1.65 V. The capacity and discharge voltage of this region depend on the competition among the three coexisting reactions [23].
Na_2_S_2_ (S) + 2Na^+^ + 2e^−^ → 2Na_2_S (S)(5)

The reduction process from Na_2_S_2_ to Na_2_S solid-solid two-phase, due to the insole ability and insulation of Na_2_S_2_ and Na_2_S, the kinetic reaction speed of this process is slow, and polarization may occur.

In general, the discharge process of room temperature sodium–sulfur batteries include the conversion of sulfur to long-chain soluble sodium polysulfide (Na_2_S_n_, 4 ≤ n ≤ 8) and the conversion of long-chain sodium polysulfide to solid Na_2_S_2_ or Na_2_S. The reaction kinetics of the formation of solid polysulfides limits the discharge efficiency, resulting in irreversible capacity loss during cycling [27]. The actual discharge capacity (1050 mAh g^−1^) of the room temperature sodium–sulfur battery is between the theoretical capacities of Na_2_S_2_ and Na_2_S, and Na_2_S_2_ and Na_2_S are the least soluble compounds in organic solvents [15], so some researchers believe that their discharge products are Na_2_S_2_ and Na_2_S [22]. Sometimes, it is also declared that the final discharge products are Na_2_S_3_ and Na_2_S_2_ [28].

## 3. Existing Problems and Solutions

Compared with high-temperature sodium-sulfur batteries, room-temperature sodium-sulfur batteries have a higher capacity. However, most reported room-temperature sodium-sulfur batteries still fail to reach one third of the theoretical capacity of sulfur [28,29,30]. This may be due to the following theoretical and technological issues:(1)Sulfur and sulfide have poor conductivity. During battery cycling, electronic conductivity is very important to the electrodes [24]. However, sulfur (5 × 10^−30^ S·cm^−1^) and its final recrystallized Na_2_S are both semiconductors, lacking inherent high electronic conductivity [31];(2)The volume expansion of sulfur causes serious changes in structure and morphology [32]. During battery discharge, the volume expansion/contraction of sulfur is a key factor in determining battery capacity. When Na_2_S_3_ is generated during the discharge process, the volume expansion rate of the sulfur cathode is 36%, for Na_2_S_2_, the volume expansion rate is 67% and reaches 157% after Na_2_S is completely generated [33];(3)Soluble polysulfide diffuses from cathode the to the anode [6,34]. The inevitable dissolution of Na_2_S_x_ (4 ≤ x ≤ 8) leads to a serious shuttle effect between the cathodes and anode, resulting in poor battery cycle stability and high self-discharge rate;(4)The formation of needle-like sodium dendrites and deposits [15]. Due to the large difference in size between sodium atoms and sodium ions, sodium easily forms unstable electrodeposited layers and dendrites. Once the dendrites reach a certain length, short circuits will occur [1,2];(5)Due to the large radius of Na^+^, the reaction activity between S and Na is slow, so the conversion of S to Na_2_S is incomplete, resulting in low sulfur utilization [35,36,37];(6)Impedance increases caused by irreversible side reactions [5,38].

Room temperature sodium–sulfur batteries face safety problems caused by the anode sodium dendrites, the insulation problem of the cathode sulfur, the shuttle effect of the intermediate product polysulfide and the loss of active materials caused by its dissolution. Starting from its cathode, electrolyte, anode and separator (Table 1 shows part of the research progress of room temperature sodium–sulfur batteries in the past six years), many methods have been developed to solve these problems:(1)Coating sulfur with conductive materials, such as carbon materials, oxides, etc., to increase the conductivity of the cathode material and speed up the battery charge and discharge process;(2)Add salt to the electrolyte, such as NaCF_3_SO_3_, NaClO_4_, NaPF_6_, etc., or introduce electrolyte film-forming additives, such as vinyl ethylene carbonate (VEC) and Na_2_S/P_2_S_5_ to improve the electrolyte;(3)Form solid electrolyte interphase (SEI) film on the surface of sodium to prevent the formation of sodium dendrites;(4)Ion-selective modification of polymer separator or β-alumina solid electrolyte separator to inhibit the shuttle effect of polysulfides.

## 4. Cathode

Sulfur is widely distributed in nature, and its content in the Earth’s crust is 0.048% by mass. Based on sulfur’s low cost, high capacity and environmental friendliness, sulfur has been extensively studied as a cathode material [18]. However, chalcogenide insulators have a conductivity of only 5 × 10^−30^ S·cm^−1^ at room temperature [61], resulting in low utilization of sulfur in the electrode. Therefore, conductive materials are required to coat sulfur particles to increase the conductivity of the cathode material. This section summarizes the research results of carbon-hosted sulfur cathodes, sulfur-organic polymer cathodes, sulfide active species cathodes, and independent binder-free cathodes reported in recent years.

### 4.1. Sulfur Cathode with Carbon as a Host

In order to reduce the shuttle effect, different low-dimensional materials are used due to the framework of the sulfur cathode, such as carbon nanospheres, graphene, porous carbon and carbon nanotubes. Their high specific surface area can inhibit the adsorption of polysulfides, and the hollow structure can alleviate the volume change during charging and discharging, which improves the battery capacity and cycle life to a certain extent.

#### 4.1.1. Sulfur-Carbon Hollow Nanospheres Composite Materials

Hollow carbon nanospheres have micro- and nanolevel hollow internal structures, and have been used in many aspects due to their large specific surface area, pore volume and low density. In the beginning, hollow carbon nanospheres, metal and metal oxides were used for lithium storage in lithium-ion batteries. The composite hollow carbon nanospheres exhibited a cycle life of at least a few hundred cycles and excellent rate performance [62,63]. Inspired by this, people began to use hollow carbon nanospheres as sulfur carriers in sodium-sulfur batteries.

The hollow carbon nanospheres have continuous interlaced C chains, and the structure is complete and compact after sulfurization, which can provide a higher tap density. When the hollow structure contains a large amount of sulfur, it can withstand the volume change of internal sulfur during battery charging and discharging. In addition, the outer shell of the carbon nanospheres can also serve as an efficient transport network and active diffusion channel for sulfur and electrons, while limiting the shuttle of polysulfides. The interconnected mesoporous carbon hollow nanospheres (iMCHS) reported by Dou et al. are used as a sulfur carrier for the cathode of a room temperature sodium–sulfur battery. The maximum reversible capacity of the battery (410 mAh g^−1^) is close to the theoretical value (418 mAh g^−1^). After 200 cycles, the reversible capacity is less than 292 mAh g^−1^, and it has excellent rate performance [48].

Modifying nanosized transition metal clusters on hollow carbon nanospheres could not only use the microporous structure of carbon spheres to fix sulfur and buffer volume changes, but also transition metal clusters can be used to enhance the conductivity and activity of sulfur and act as electrocatalysts. Quickly reduce polysulfides to short-chain sulfides, thereby improving the electrochemical performance of RT-Na/S batteries [55]. The hollow carbon nanosphere electrodes modified by transition metal Fe, Cu, Ni and Co nanoclusters have effectively improved the battery performance [64]. In-situ Raman spectroscopy, synchrotron X-ray diffraction and density functional theory have confirmed Fe and Cu. The electrocatalysis of Ni and Co effectively hinders the shuttle effect. Recently, Wang et al. [65] modified gold nanodots on layered N-doped carbon microspheres (CN/Au/S) as sulfur cathodes, which improved sulfur utilization, cycle stability and rate performance. Catalyzes the low kinetic process of Na_2_S_4_ into Na_2_S_2_ (discharge process) or S (charge process), and realizes a completely reversible conversion reaction at the S cathode. Now someone has designed a template method for preparing metal/carbon–sulfur carriers [66], which can be used in liquid/solid room temperature Na/S batteries. The addition of transition metals provides more possibilities for electrodes in sodium ion batteries and other batteries.

In addition to transition metals, transition metal sulfides can also be modified in medium carbon nanostructures. The porous carbon structure can effectively buffer the volume change during the charge and discharge process and store S_8_, and the polysulfide sites enhance the catalysis of multistep S conversion, effectively inhibit the dissolution of long-chain polysulfides. Improved the kinetics of short-chain polysulfide conversion. For example, the electrocatalytic sulfur cathode made of porous core-shell structure and ZnS or CoS_2_ has achieved high cycle performance with a reversible capacity of 570 mAh g^−1^ even after 0.2 A g^−1^ cycles over 1000 cycles [67]. FeS_2_ is grown in situ in a carbon nanocage as a cathode. FeS_2_ nanoparticles have a lower Na^+^ diffusion barrier, a strong binding energy and a high affinity for sodium polysulfide. The cathode provides a high sulfur content of 65.5 wt %, high reversible capacity and excellent cycle stability [68]. The combination of transition metal polysulfide and porous carbon structure is undoubtedly an ideal sulfur host for fixing polysulfide and realizing the reversible conversion of polysulfide to Na_2_S.

#### 4.1.2. Sulfur-Graphene Composite Cathode

The characteristics of high theoretical surface area, elastic modulus and thermal conductivity/conductivity make graphene (GO) an ideal sulfur host material [17]. Early sulfur-graphene composite cathodes were used in lithium-sulfur batteries, and polysulfides were physically restricted by changing the pore structure of graphene [69]. Later, it was discovered that graphene oxide (rGO)-sulfur composite cathodes can chemically adsorb sulfur, but the conductivity of GO depends on its degree of oxidation, which has certain limitations [70]. Further use of N, S doping to prevent sulfur loss [71,72]. In 2017, Ghosh et al. [73] proposed a cathode structure (CS90-rGO) with a sulfur loading of about 90 wt % for room temperature sodium-sulfur batteries. In the presence of elemental sulfur (CS-90), benzoxazine undergoes thermal ring-opening polymerization to generate sulfur polymer, which is then compounded by reducing graphene oxide. Plasticizable organic sulfide cells and extremely high conductivity rGO(S) solve the problems of non-deposition of final discharge products and low conductivity of active materials, so the CS90-rGO electrode structure has excellent Coulombic efficiency (99%) and excellent cycle life.

Later, they synthesized an independent cathode composed of rGO, S nanoparticles, mixed valence manganese oxide (Mn_x_O_y_) and sodium alginate/polyaniline hybrid binder through the bottom-up method [74]. Among them, rGO provides excellent stability and restricts the movement of active sulfide ions; S nanoparticles can promote the conversion reaction at room temperature; manganese oxide can fix polysulfides; sodium alginate/polyaniline hybrid matrix is used as the electron/ion conductivity of the cathode. The components cooperate with each other, and the room temperature sodium–sulfur battery using the cathode has a specific weight capacity of 737 Wh kg^−1^ after two cycles, and the capacity remains at 660 W h kg^−1^ after 50 cycles, with excellent cycle and rate performance.

In addition to nano sulfur can promote the conversion of polysulfides, metal oxides also have excellent catalytic properties. Du et al. [16] used the high conductivity of reduced graphene oxide and the catalytic effect of VO_2_ on the conversion of soluble polysulfides to solid sulfides, and a three-dimensional layered cathode with VO_2_ nanoflowers grown in situ on reduced graphene oxide (rGO) was designed and prepared. This structure can not only greatly improve the conductivity of the sulfur cathode, but also accelerate the conversion of polysulfides, alleviate the problem of polysulfide dissolution, and enhance the cycle capacity. The experimental results show that the first-turn reversible capacity of the composite cathode is 876.4 mAh g^−1^ at 0.2 C, and it is cycled 1000 times at 2 C, and the capacity attenuation is only 0.07% per turn, showing a high capacity and excellent cycle stability.

#### 4.1.3. Sulfur-Porous Carbon Composite Cathode

Porous carbon has the properties of large specific surface area, light weight, high chemical stability, electrical and thermal conductivity, and can be used as energy storage and electrode materials. Due to the high affinity of carbon to sulfur, in sodium–sulfur batteries, the compound of porous carbon and sulfur forms a sulfur-porous carbon cathode, which plays a role of fixing sulfur to control the shuttle effect of the battery, thereby improving battery performance. The following introduces microporous carbon (pore size 0.35–2 nm), mesoporous carbon (pore size 2–50 nm) and graded porous carbon.

(1) Sulfur-microporous carbon composites

Microporous carbon material is the best choice for physical sulfur fixation [34]. Wei et al. reported a microporous carbon polyhedral sulfur composite (MCPS) cathode synthesized with molecular sieve type MOF (ZIF-8). The synthesized microporous carbon polyhedron has a uniform sponge-like microporous structure with a high BET specific surface area of 833 m^2^ g^−1^. After carbonization and sulfur treatment, the morphology and structure of the material still remain the rhombic dodecahedron of ZIF-8. MCPS cathode combined with ionic liquid electrolyte, the assembled sodium–sulfur battery has excellent cycle performance. At higher current density, the sulfur content in the cathode is relatively high (46%), and the coulombic efficiency is close to 100%.

In 2017, Pint et al. [31] used sucrose as the sulfur carrier to synthesize microporous carbon with a sulfur loading of 35 wt %. The battery showed good cycle performance and capacity, and the coulombic efficiency was higher than 98%. In the same year, Hu et al. [75] prepared microporous carbon by carbonizing polyvinylidene fluoride (PVDF) as a precursor, and successfully obtained micropores with a larger volume (0.472 cm^3^ g^−1^) and a smaller pore size (<0.7 nm). The microporous carbon/sulfur composite material shows stable cycle performance and close to 100% coulombic efficiency in RT Na/S battery.

In addition to simply using microporous carbon-rich pores for physical sulfur capture, its sulfur capture capability can also be further improved by doping. Mou et al. [76] prepared a nitrogen self-doped porous carbon (NPC) sheet through simple self-assembly. Through the large specific surface area, rich microporous structure and the synergistic effect of N doping, polysulfides are restrained from both physical and chemical aspects to ease the dissolution of polysulfides. The N-doped porous carbon–sulfur cathode shows superior electrochemical performance, and the reversible capacity after 400 cycles at 0.5 C is 418.9 mAh g^−1^.

(2) Sulfur-mesoporous carbon composites

Zheng et al. [42] reported a reasonable design of nano-copper-assisted immobilization of S in the mesoporous carbon (HSMC) cathode with high specific surface area. After 110 cycles, the capacity retention rate is still as high as 610 mAh g^−1^. The HSMC-Cu-S cathode is the earliest long-life RT Na-S battery cathode with a higher S load (50%).

(3) Sulfur-hierarchical porous carbon composites

The sodium polysulfide in the carbon pores can be effectively captured by the electrostatic interaction between the pore wall and the sulfur and nitrogen-sodium (Na-N) bonds. Qiang et al. [49] synthesize hierarchical porous carbon (N, S-HPC) with high concentration of nitrogen and sulfur from low-cost raw materials. The larger concentration of heteroatoms in the carbon skeleton can pass through the static electricity between the pore wall and sulfur and Na-N bonds. The interaction effectively captures the sodium polysulfide in the nanopores, thereby inhibiting the side reaction between the polysulfide and the carbonate electrolyte, so as to improve the cycle stability and charge-discharge cycle performance of the room temperature Na-S battery. In addition, the effect is still effective in the TEGDME electrolyte.

Recently, Guo et al. [60] produced a porous material with ultramicropores as the main body. The material is obtained by carbonizing and activating coffee grounds. As a sulfur carrier, the elemental sulfur enters the ultramicropores in the form of S_2-4_, and the loading amount can reach 40 wt %. Through in-situ UV testing and density functional theory calculations. Facts have proved that during charging and discharging, only Na_2_S is formed and the formation of polysulfides is suppressed. Utilizing the micropore restriction, a one-step reaction from elemental sulfur to Na_2_S can be realized. In the RT Na-S battery, the electrode has a daily capacity decay rate of only 0.17%, showing extremely low self-discharge. At a current density of 0.1 C, a high reversible specific capacity of 1492 mAh g^−1^ and a long cycle life are obtained. After 2000 cycles, there was almost no capacity degradation and excellent cycle stability (Figure 5a).

#### 4.1.4. Sulfur-Carbon Nanotube Composite Cathode

The uniform porous structure of the conductive matrix and the good distribution of the sulfur active material in the conductive matrix are two major factors that affect the battery performance. Carbon nanotubes (CNTs) have excellent flexibility and conductivity, so that their combination with active materials can improve the overall conductivity of the electrode. Using the solubility of long-chain sodium polysulfide. In addition to the above-mentioned sulfur hosts, porous carbon nanotubes are also a multifunctional sulfur host. In order to solve the problem of rapid capacity decay and low reversible capacity of room temperature sodium–sulfur batteries, researchers at the University of Wollongong in Australia have developed a nanomaterial with nickel sulfide nanocrystals implanted in nitrogen-doped porous carbon nanotubes for sodium-sulfur batteries cathode [60]. The material is not only superior in performance, but also suitable for mass production and commercialization. Nitrogen-doped porous carbon nanotubes implanted with nickel sulfide nanocrystals are a multifunctional sulfur host. It is found that the carbon backbone in the host can provide a shorter ion diffusion path and a fast ion transfer rate. The doped nitrogen sites and the polar surface of nickel sulfide can improve the adsorption capacity of polysulfides and provide strong catalytic activity for the oxidation of polysulfides, indicating that sodium–sulfur batteries can have longer cycle life, high performance, and quick charge and discharge. The researchers said that the next step is to expand the production of this material, push sodium-sulfur batteries from the laboratory to industry, and put this battery system into practical applications.

Yu et al. [43] proposed a reversible Na/dissolved sodium polysulfide battery, using a binder-free multiwalled carbon nanotube (MWCNT) fabric electrode. Liquid-phase sodium polysulfide as a cathode can easily disperse and evenly distribute the sulfur active material into the conductive matrix. The liquid sodium polysulfide/MWCNT fabric cathode provides excellent sulfur active material utilization and capacity retention. However, the battery capacity continues to decrease after the first cycle, and the capacity decline is mainly caused by the poor reversibility of low-order polysulfides. With the intermediate product (polysulfide) as the starting cathode, it is not affected by the conversion of sulfur to long-chain sodium polysulfide, and the electrochemical characteristics of the battery in the low-voltage platform area can be studied.

The carbon material with high specific surface area and pore structure can be used as a host for sulfur loading. The addition of modified metals and their sulfides not only effectively inhibits the shuttle effect, but also improves electrode conductivity and catalyzes the conversion of polysulfides. The purpose of improving battery performance is achieved. However, their pore volume still needs to be further increased to accommodate more sulfur and further improve the cycle performance of RT Na-S batteries, so as to achieve commercial large-scale applications.

### 4.2. Sulfur-Organic Polymer Cathode

The sulfur-organic polymer cathode mainly uses the reaction of sulfur and organic polymer to achieve the purpose of inhibiting the production of long-chain polysulfides. The current research results in this area mainly include various forms and carbonized sulfurized polyacrylonitrile (S-PAN) composite cathodes and covalent sulfur-based carbon (C) cathodes.

#### 4.2.1. S-PAN Composite Cathode

In 2002, Yoshimoto et al. [77] reported on vulcanized polyacrylonitrile (S-PAN) composite material as the cathode material for Li/S batteries. Five years later, Wang et al. [29] used S-PAN as the cathode of RT Na/S battery for the first time. S is bonded to the carbocyclic ring and exists between two adjacent polypyridine rings as an S_2-3_ chain, effectively avoiding the formation of polysulfides. During the discharge process, the S molecules on the S-PAN framework can be completely converted into Na_2_S, and it has a high initial discharge capacity of 654.8 mAh g^−1^, a charge-discharge efficiency of about 100%, and a sulfur utilization rate of 87%.

Hwang et al. [32] prepared and carbonized vulcanized PAN nanofibers by electrospinning. The sulfur atoms in the carbonized vulcanized polyacrylonitrile are firmly bound in the turbocharged carbon matrix, which greatly improves the stability of the battery. In 300–500 cycles, the coulombic efficiency was as high as 99.84%. Figure 5b shows the structural changes of PAN during the vulcanization and carbonization process. First, the cyclization of polyacrylonitrile is carried out, and then the sulfur radicals formed by the cracking of S_8_ react with the carbon matrix derived from polyacrylonitrile to form the final carbon–sulfur composite structure. The arrangement of sulfur atoms in the carbon matrix is the key to suppressing polysulfur and improving battery stability. Zhu et al. [57] mixed polyacrylonitrile and sulfur by heating for carbonization to synthesize S/CPAN composite material as the cathode of all solid-state room temperature sodium–sulfur batteries. S_8_ promotes the cleavage and cyclization reaction of C≡N. S_8_ splits into short-chain Sn (*n* ≤ 4) to react with the polymer main chain, and bind to the polymer main chain in the form of chemical bonds to dehydrogenate the material to H_2_S. There is almost no S in this composite cathode. Since the battery is all solid, the dynamics is slower than that of a liquid electrolyte battery. The second volume attenuation is attributed to the irreversible capacity provided by S and the irreversible sodium insertion of the carbon framework. The maximum discharge capacity of the all-solid-state battery is 274 mAh g^−1^, and it still maintains 251 mAh g^−1^ after 100 cycles, and the coulombic efficiency is close to 100%.

Later, Kim et al. [46] prepared a flexible S-PAN fiber web without a binder. The S-PAN fiber web can be easily restored after being bent 180°, and it has an excellent flexibility, and stability. After the second discharge, the fiber web has no foreign particles and defects. A sodium–sulfur battery with S-PAN fiber mesh as the cathode can still obtain a specific capacity of 257 mAh g^−1^ after being cycled 200 times at 0.1 C. Li et al. [56] prepared composite cathode Te_0.04_S_0.96_@pPAN by doping Te in vulcanized polyacrylonitrile as a eutectic accelerator, which accelerated the reaction kinetics rate, improved the conductivity of electrons, and made the sulfur more fully utilized. As a result, the performance of RT Na-S battery has been greatly improved. At the same time, the composite cathode can be compatible with carbonate electrolyte or ether electrolyte. Te_0.04_S_0.96_@pPAN provides 1236 mAh g^−1^ and 1111 mAh g^−1^ in carbonate and ether electrolytes at 0.1 A g^−1^, respectively.

The original S-PAN cathode mainly uses the one-step reaction of S on its skeleton to directly turn into Na_2_S to reduce the generation of long-chain polysulfides. The carbonized S-PAN cathode has different chemical bonds due to different preparation conditions, so there are also differences in capacity. The prepared fibrous S-PAN has excellent flexibility, and there is no defect on the cathode surface after the second discharge, indicating that the battery is very stable. The eutectic accelerator can accelerate the reaction kinetics, and Te_0.04_S_0.96_@pPAN composite shows an excellent specific capacity.

#### 4.2.2. Covalent Sulfur-Based Carbonaceous (CSCM) Cathode

Covalent sulfur can effectively avoid the formation of sodium polysulfide, thereby avoiding its dissolution in the electrolyte. In addition, the covalent sulfur in the carbonaceous material is reversible during the charge and discharge process, which is conducive to long-term cycling. Fan et al. [47] used pure benzo[1,2-b:4,5-b’] dithiophene-4,8-dione (BDTD) sulfur powder to synthesize covalent sulfur-based carbonaceous (CSCM) cathode by pyrolysis. Most of the sulfur in the CSCM material is in the form of covalent bonds. The results of exploring the sodium storage mechanism of CSCM electrodes show that the CSCM high platform capacity exceeding 0.6 V accounts for more than 75% of the total capacity, which indicates that the capacity of carbon-based materials mainly comes from the reaction of covalent sulfur and Na^+^. During the discharge process of CSCM, there is no obvious plateau at 2.0–2.2 V, indicating that Na_2_S_n_ (4 ≤ *n* ≤ 8) is hardly formed during the cycle, which avoids the reduction of capacity. In addition, it is inferred that the sodium storage mechanism of the CSCM electrode, during the sodium insertion process, covalent sulfur may react with Na ^+^ to form Na_2_S. In the process of sodium removal, S^2−^ may return to covalent sulfur. For the CV curve, the two negative peaks of 0.1 V and 2.0 V (Figure 5c) were attributed to the loss of Na^+^ from disordered carbon and the extraction of Na^+^ from covalent sulfur, respectively. The room temperature sodium-sulfur battery assembled with CSCM cathode had a high reversible capacity above 1000 mAh g^−1^, a long cycle stability of 900 cycles, a low capacity attenuation of 0.053% per cycle and an excellent coulombic efficiency close to 100%.

The sulfurized polyacrylonitrile composite cathode mainly uses the bonding of sulfur and carbon ring to achieve sulfur fixation to reduce the shuttle effect. The covalent sulfur-based carbonaceous cathodes can be combined with sodium and can be separated during the deposition and stripping process of sodium. The bond is changed back to covalent sulfur, which provides a good reversibility for the battery. The sulfur-organic polymer positive electrode improves the electrochemical performance through different mechanisms, such as the cycle stability of the battery.

### 4.3. Metal Sulfide Cathode

Although the hollow carbon nanospheres, graphene, porous carbon and carbon nanotubes used as the sulfur main body have a large specific surface area, they lack a catalytic effect. At present, metal sulfides are used as cathodes instead of sulfur cathodes, by catalyzing the conversion of sulfur and polysulfides and using S-sites to store Na^+^, so as to reduce the shuttle effect and improve the electrochemical performance of the battery.

#### 4.3.1. Sodium Sulfide

The safety and energy of the all-inorganic solid-state sodium–sulfur battery are relatively high, but the insufficient three-phase contact between the sulfur active material, the solid electrolyte and the conductive carbon will result in greater resistance at the electrode/electrolyte interface. In order to solve this problem, Yue et al. [78] studied a Na_3_PS_4_-Na_2_S-C nanocomposite cathode for inorganic solid sodium-sulfur batteries. The uniform distribution of Na_3_PS_4_ and Na_2_S in the nanoscale carbon matrix ensures sufficient interface contact and mixed conductivity. Na_3_PS_4_ can be used as both a solid electrolyte and an active cathode material. Since the charge transfer reaction requires only two-phase contact, the electrode/electrolyte interface contact has inherent advantages. The nano composite cathode has a high initial discharge capacity of 869.2 mAh g^−1^ at 50 mA g^−1^, and the battery has better cycle performance and rate performance than other all-inorganic solid sodium–sulfur batteries.

#### 4.3.2. Bismuth Sulfide

Li et al. [79] prepared graphite nanosheets coated with Bi_2_S_3_ composite materials by the precipitation method and ball milling method. The amorphous carbon coated on Bi_2_S_3_ particles has two functions, one is to increase electrical conductivity, and the other is to act as a protective layer to prevent the dissolution of polysulfides. The sodium storage mechanism of Bi_2_S_3_ is that during the discharge process, Bi_2_S_3_ decomposes into Bi and S, and S and Na form Na_2_S. Bi does not undergo an alloying reaction with Na, but undergoes a Na^+^ insertion process. The initial capacity of Bi_2_S_3_@C as the cathode of sodium-sulfur battery can reach 550 mAh g^−1^. After 100 cycles, the capacity retention rate is 69%.

#### 4.3.3. Molybdenum Sulfide

Ye et al. [80,81] proposed the concept of sulfur equivalent cathode material, replacing elemental sulfur with sulfide. Amorphous chain MoS_3_ as the cathode material of room temperature sodium-sulfur battery can quickly absorb Na^+^. During repeated sodium insertion and removal, the Mo-S bond is not broken, and metallic molybdenum, Na_2_S and polysulfides will not be produced. It fundamentally inhibits the formation of polysulfides, and has excellent capacity and long cycle life.

Wang et al. [82] used amorphous chain α-MoS_5.7_ in lithium-ion batteries. Not only can the rapid diffusion of ions be realized, but also the storage of Li ^+^ on the S-site can be realized. The shuttle effect can be prevented in both sodium-sulfur batteries and lithium–sulfur batteries. Since MoS_x_ can store alkali metal ions at the S-site, a further increase in the S content will also increase the capacity of the cathode. Amorphous non-crystalline chain MoS_5.6_ is also an excellent cathode material for sodium metal batteries. 50 mA g^−1^ MoS_5.6_ has an excellent capacity of 537 mAh g^−1^. Cyclic voltammetry shows that the peak value of Na/Na^+^ is about 1.8 V, which is similar to S, but there is almost no sign of polysulfide formation or shuttle [83]. In summary, sulfur-rich molybdenum polysulfide is a promising alternative to sulfur cathode materials.

#### 4.3.4. Cobalt Sulfide

Cobalt sulfide has a large specific surface area, can accommodate sodium polysulfide, withstand volume expansion and has a catalytic effect on the formation of polysulfide. Aslam et al. [59] designed a hollow, polar and catalytically active cobalt sulfide double prism as a high-efficiency sulfur host for sodium-sulfur batteries. It can capture polysulfide polysulfides and catalyze the conversion of long-chain sulfides to short-chain solid sulfides. The bipyramid prisms sulfur@cobalt sulfide composite brings excellent electrochemical performance to the sodium-sulfur battery. The second discharge specific capacity is 755 mAh g^−1^ at a current of 0.5 C. After 800 cycles, it still has 675 mAh g^−1^. Specific capacity, capacity attenuation rate is as low as 0.0126%

The main considerations for the design of the room temperature sodium–sulfur battery cathode are the following: excellent electronic conductivity, small electrode polarization, large electrode material porosity, good elasticity, good conductivity, large sulfur loading and the volume change during battery charging and discharging. The sulfur hosts cathode uses the porous structure of the material to adsorb and fix sulfur. The sulfur-organic polymer cathode suppresses the formation of polysulfides through chemical bonds. The independent binder-free cathode avoids the disadvantages of the binder itself, and improves the conductivity so that the electrochemical reaction is more reversible.

## 5. Electrolyte

Electrolyte is an important part of the battery and is closely related to the cycle efficiency, cycle life and safety of the battery. Sodium-sulfur battery electrolyte must meet the conventional requirements of ionic conductivity, electronic insulation, thermal stability, chemical stability, electrochemical stability, excellent wettability of the electrode, environmental friendliness and low cost. Moreover, it has no reactivity to sodium and has high solubility to polysulfides. The electrolyte used in room temperature sodium-sulfur batteries can be classified into liquid, gel and solid electrolytes. This section mainly discusses the application of ether, carbonate, ionic liquid electrolyte and gel electrolyte in liquid electrolyte in room temperature lithium-sulfur batteries.

### 5.1. Liquid Electrolyte

Liquid electrolyte is currently the most common electrolyte used in sodium-sulfur batteries, and is mainly classified into ether-based, carbonate-based and ionic liquid electrolytes.

#### 5.1.1. Ether-Based Electrolyte

NaCF_3_SO_3_ is dissolved in a solvent based on TEGDME as a liquid electrolyte for Na/S batteries [28]. Although compared with the polymer electrolyte, the conductivity of the electrolyte has been enhanced, but after 10 cycles, the discharge capacity was severely reduced from 538 to 240 mAh g^−1^. Yu et al. [27] proposed a RT-Na-S battery operated by sulfur/long-chain sodium polysulfide redox couple. By inserting a nano-carbon foam layer between the sulfur cathode and the separator to locate the soluble polysulfide and prevent it from migrating to the anode, an advanced cathode structure is formed. Sodium sulfur is obtained by dissolving NaClO_4_ and NaNO_3_ in TEGDME. The reversible sulfur/long-chain sodium polysulfide battery can provide a stable output energy density of 450 Wh kg^−1^ and a very low energy cost. TEGDME is stable in the presence of sodium sulfide [45,84], but polysulfides are easily soluble in TEGDME, so it is important to inhibit the shuttle of polysulfides. In addition to the commonly used TEGDME for ether-based electrolytes, there are also electrolyte solutions prepared by dissolving NaCF_3_SO_3_ in dry 1,3-dioxolane (DOL)/dimethoxyethane (DME) [19].

#### 5.1.2. Carbonate-Based Electrolyte

Mix ethylene carbonate (EC) and dimethyl carbonate (DMC) in a weight ratio of 2:1, and then dissolve NaClO_4_ in a solvent to obtain NaClO_4_EC-DMC electrolyte, which is the electrolyte used in earlier room temperature sodium-sulfur batteries liquid [29]. During the first discharge, the battery has a specific capacity of 654.8 mAh g^−1^, and the reversible specific capacity of the sulfur composite cathode is about 500 mAh g^−1^, and it remains stable in the subsequent cycles, with a charge-discharge efficiency of about 100%. The average charge and discharge voltages were 1.8 V and 1.4 V, respectively. The carbonate-based electrolyte is combined with a small sulfur molecule cathode [40], and the battery exhibits high electrochemical activity. Na can be completely reduced to Na_2_S, and it has a stable circulation capacity in the electrolyte. The battery has good cycle stability (reversible capacity 1000 mAh g^−1^) and a service life of 200 cycles.

#### 5.1.3. Ionic Liquid Electrolyte

Since ionic liquids have the characteristics of thermal stability, non-flammability, stable performance and low volatility, compared with other organic electrolytes, ionic liquid electrolytes have the following advantages: wide application range, high thermochemical stability and not easy to volatilize during circulation. However, the conductivity of ionic liquid batteries is limited, and its discharge capacity and rate performance are not as good as organic electrolyte batteries. When researchers used ionic liquid electrolyte for lithium ion batteries, the battery showed excellent cycle performance and high rate performance [85]. The electrolyte based on propylpiperidine has higher conductivity and effectively inhibits battery short circuit. The life of these batteries is extended by nearly 10,000 times. In 2016, 1-methyl 3-propyl imidazole chloride salt ionic liquid electrolyte (SiO_2_-IL-ClO_4_) with silica nanoparticles as an additive was used in room temperature sodium–sulfur batteries [34]. The battery uses a microporous carbon-sulfur composite cathode. It has been proven that the Na-S battery of this design can achieve excellent cycle performance. At a higher current density, the sulfur content in the cathode is relatively high, and the coulombic efficiency is close to 100%. Studies have shown that ionic liquid electrolytes are an effective substitute for unstable and flammable conventional carbonates, and can be used to prepare electrolytes for future sodium-sulfur batteries.

Although the liquid electrolyte is convenient and widely used, it still has many problems, such as easy explosion, easy leakage, providing a habitat for soluble long-chain sulfides and easy generation of sodium dendrites.

### 5.2. Gel Electrolyte

In the published work, the electrolyte used in the first room temperature sodium-sulfur battery is a gel electrolyte. In the early study, Park et al. [14] prepared PVDF gel polymer electrolyte with tetraethylene glycol dicarboxylate plasticizer and NaCF_3_SO_3_ and tested its applicability as a sodium–sulfur battery electrolyte. The initial discharge capacity of the solid-state sodium-sulfur battery using the PVDF gel polymer electrolyte at room temperature is 489 mAh g^−1^, two plateau potential regions of 2.28 V and 1.28 V appear, and the capacity decays to 40 mAh g^−1^ after 20 cycles. In 2008, Kim et al. [86] prepared a PVDF gel polymer electrolyte containing tetraethoxy plasticizer and NaCF_3_SO_3_. As a result, its performance was not improved, and the capacity dropped to 36 mAh g^−1^ after 20 cycles.

Later, silica nanoparticles were dispersed in polyvinylidene fluoride-hexafluoropropylene copolymer (PVDF-HFP) to prepare a sodium ion conductive gel polymer electrolyte [87]. The nanocomposite gel material is dimensionally stable, at room temperature, the highest ionic conductivity of the electrolyte is 4.1 × 10^−3^ S·cm^−1^ and the dispersion of SiO_2_ significantly increases the sodium ion transport number to the maximum. The FTIR study confirmed that due to the encapsulation of the NaTF-EC-PC solution, the gel interaction between the ions and the filler and the polymer gel and the crystal structure of the main polymer PVDF-HFP might undergo conformational changes. The prototype of the sodium-sulfur battery made with the optimized gel electrolyte has a first discharge capacity of about 165 mAh g^−1^, and the capacity declines sharply afterwards, possibly due to the formation of irreversible sodium polysulfide during the charging process.

Kumar et al. [88] integrated ionic liquid and gel, and carried out a process of preparing sodium carbonate-free ion conductive gel polymerization by coating the polyvinylidene fluoride-hexafluoropropyl polymer matrix with sodium trifluoride in ionic liquid. It was confirmed that the coating of NaCF_3_SO_3_/EMITf solution in PVDF-HFP polymer significantly reduced the crystallinity of the host polymer. The gel polymer electrolyte separator has extremely low impedance (<20 Ω) and high conductivity of 5.7 × 10^−3^ S·cm^−1^ at room temperature. Its independent mechanical properties and dimensional stability make it an effective substitute for unstable and flammable traditional carbonate electrolytes.

Due to the dimensional stability of the gel electrolyte, a lot of research has been carried out on it, and it is found that a single gel electrolyte cannot improve the electrochemical performance of the battery well. During the cycle, the battery capacity drops sharply due to the irreversible conversion of polysulfides. However, it can be modified, such as adding an ionic liquid to obtain a low-impedance, high-conductivity electrolyte.

### 5.3. Solid Electrolyte

In order to deal with the problems of flammability, leakage and dendrite growth in liquid electrolytes, solid polymers have been proposed. The solid polymer can replace the cathode and anode while ensuring the transmission of sodium ions [83]. According to the composition, the solid electrolyte of sodium-sulfur battery can be sorted into inorganic electrolyte and polymer electrolyte [89].

#### 5.3.1. Inorganic Solid Electrolyte

Inorganic solid electrolytes include Na-β-Al_2_O_3_, NASICON type, sulfur compounds and borohydrides. An et al. [90] used ionic liquid *N*-butyl-*N*-methylpyrrolidinium bis(fluorosulfonyl) imide (Pyr_14_FSI) to modify the anode/electrolyte interface. Through characterization and electrochemical experiments, the stability of the in-situ formation of SEI and electrolyte/anode interface was proved under static and dynamic conditions.

#### 5.3.2. Polymer Solid Electrolyte

Yu et al. [91] used a NASICON-type Na^+^ solid electrolyte separator and used Na_3_Zr_2_Si_2_PO_12_ as a separator to block polysulfides. The interface and the PIN coating can provide an elastic buffer layer between the Na anode and the Na_3_Zr_2_Si_2_PO_12_ solid electrolyte, greatly reducing the risk of solid polymer particles breaking. The cycle stability of Na-S battery with a solid electrolyte separator inserted by PIN is improved. For PEO-NaCF_3_SO_3_, the addition of MIL-53 (Al) improves the ion conductivity and the number of sodium ion transfer. Ge et al. [92] prepared a flexible PEO-NaCF_3_SO_3_-MIL-53 (Al) solid electrolyte. The all-solid sodium-sulfur battery assembled with NaCF_3_SO_3_-MIL-53 (Al) electrolyte has higher capacity and better cycle stability. Inorganic solid electrolyte has higher polymer solid electrolyte, higher ionic conductivity, better mechanical strength and thermal stability, but the interface resistance between polymer electrolyte and electrode material is higher, which is not conducive to substance diffusion.

## 6. Separator

Although liquid sodium-sulfur batteries have potential safety hazards, liquid electrolytes have a shuttle effect and rapid battery failure, liquid electrolytes are still the most widely used electrolyte in room-temperature sodium-sulfur batteries [93]. In order to guide sodium ions and prevent polysulfides from migrating to the anode, the liquid electrolyte should be used with the battery separator to ensure the battery’s cycle performance [2]. At present, the separators often used in room temperature sodium-sulfur batteries can be roughly sorted into three categories. One is glass fiber separators and polyolefin separators made of polypropylene or polyethylene. This type of separator is the most widely used; the other is Nafion separators and premodified Nafion functional separators. This type of separator has excellent performance in suppressing the shuttle effect; there is another type of separator that modifies the intermediate layer of the conventional separator. This type of separator has few research results and is still in the development stage. The original sodium-sulfur battery separator is β-Al_2_O_3_, which can be used as both a solid electrolyte and a separator. However, this sodium-sulfur battery needs to work at a temperature higher than 300 °C. The molten sodium and sulfur may react explosively, and β-Al_2_O_3_ cannot prevent the shuttle effect of polysulfides [94]. Later, glass fiber separators and polyolefin separators were widely used, but the binding energy between the bond and Na_2_S/polysulfide was low, so polysulfide inevitably migrated to the anode [45,54,95]. The Nafion functional separator and carbon-based intermediate layer have the advantages of ion selectivity and other advantages, and they perform better in suppressing the shuttle effect.

### 6.1. Glass Fiber Separator

The large porosity and thickness of the glass fiber separator require a large amount of liquid electrolyte to fill the pores of the separator, which will not only reduce the energy density of the battery and increase the risk of flammability, but also even dissolve and shuttle intermediate products. At the same time, sodium dendrites can easily pass through with its large pore size, which can also cause safety problems such as short circuits. These problems with glass fiber separators reduce the cycle life and coulomb efficiency of sodium–sulfur batteries and hinder the development of sodium–sulfur batteries [96]. The use of glass fiber separator alone is not superior, but it can also contribute to the hindrance of polysulfides by modifying it. Li et al. [97] simultaneously coated graphene and Fe^3+^/polyacrylamide nanospheres (FPNs) on the separator through vacuum filtration to form a FPNs-graphene functionalized separator (FPNs-G/separator). Utilizing the strong polarity of Lewis acid-base, FPNs chelate with adsorbed polysulfides, chemically inhibiting the shuttle effect; the introduction of graphene not only improves the mechanical fragility of FPNs, but also physically hinders polysulfides. The cathode mixed with mesoporous nitrogen-doped carbon nanometers and sulfur obtained by FPNs-G/separator carbonization of FPNs shows a higher discharge capacity (639 mAh g^−1^ at 0.1 C) and a longer cycle life (800 cycles). The rear capacity is retained at 68.3% at 0.5 C).

### 6.2. Polyolefin Separator

Polyolefin separators are widely used in various metal batteries due to their low cost, lightness and thinness, good mechanical strength and toughness. Compared with glass fiber separators, polyolefin separators require less electrolyte, which reduces the dissolution and shuttle of polysulfides and extends the cycle life of the battery. The polyolefin separator alone still cannot achieve the expected effect of the ideal separator, so the polypropylene separator was modified. Zhou et al. [98] used the “graft-filtration” strategy to chemically combine the sodium ion conductive polymer layer with the polyolefin separator, and developed the “single sodium ion conductive polymer graft side|functional low-dimensional material coating side” Janus separator (Figure 6a). The single ion conductive PMTFSI Na graft layer can effectively improve the wettability of the electrolyte, and the DN-MXene coating layer can effectively catalyze the conversion of polysulfides. In general, the Janus separator can significantly improve the reversible capacity and cycle life of the battery.

### 6.3. Nafion Separator

Manthiram et al. [99] compared polyolefin separator with Nafion separator. Compared with the polyolefin separator, the initial cycle capacity of the room temperature sodium-sulfur battery was increased from about 500 to 750 mAh g^−1^ when the Nafion separator was used. Studies have proved that Nafion separator can selectively transport cations, and when used in sodium-sulfur batteries, polysulfides can be effectively confined to the cathode area. Based on the expensive price and low ionic conductivity of Nafion separators, people began to explore better separators from the premodification of Nafion separators. Kaskel et al. [41] proposed for the first time a room temperature sodium–sulfur battery using a polypropylene separator with Nafion coating. The separator improves the utilization rate of sulfur and the conductivity of sodium ions, and reduces the penetration of polysulfides. The capacity fluctuation is small and the performance is stable during the charge and discharge cycle. In 2017, in order to improve the sulfur utilization rate and the pretreatment capacity of the battery, the Manthiram team proposed a Na||Na-Nafion/AC-CNF||Na_2_S/AC-CNF battery [50]. The combination of Na-Nafion and AC-CNF (activated carbon-carbon nanofiber) further improves the sulfur utilization rate of the battery, and at the same time provides a secondary current collector that reuses polysulfides in an electrochemical manner. The research explored Nafion separator suppression. The shuttle effect is mainly realized by the “structural effect” and “electronic effect” of hydrophilic pores smaller than 5 nm under the negative charge environment of the Nafion separator. In addition, the Al_2_O_3_-Nafion separator can obtain a higher capacity retention rate, and after 100 cycles, it still maintains a capacity of about 250 mAh g^−1^ [100].

### 6.4. Intermediate Layer

In addition to the above-mentioned separator, an intermediate layer can also be added between the separator and the cathode according to the required properties of the battery. Carbon nanofoam is a kind of porous conductive material. Yu et al. [27] put a carbon nanofoam interlayer (CNF) with a nanosized solid frame and pore space between the glass fiber separator and the sulfur cathode (Figure 6b). This structure is similar to the sulfur cathode. With fine surface contact, resistance can be reduced. It can also suppress the shuttle effect and act as a secondary current collector. In addition, the interlayer is well digested to capture the active material and the volume change of sulfur during discharge. In the first cycle or the first few cycles, the battery capacity will drop sharply regardless of whether the interlayer is present or not. After the CNF intermediate layer is soaked in the electrolyte for a complete cycle, the discharge voltage rises to the normal value, and there is no change in the shape of the discharge curve.

In summary, in order to obtain a room temperature sodium–sulfur battery with stable cycle performance and long life, the most important task of the separator is to guide the migration of Na^+^ and inhibit the shuttle of polysulfides. Sodium polysulfide dissolved in the electrolyte must pass through the separator to reach the anode. In other words, the separator should only allow solvated Na^+^ to pass through and block long-chain polysulfides. Therefore, the ideal separator should have high ion selectivity and physical adsorption, which can be achieved by surface modification of the separator or interlayer.

## 7. Anode

In addition to the dissolution of polysulfides, sodium-sulfur batteries also have some difficult problems on the anode. Metal sodium is an excellent electrical conductor, and its corrosion resistance and strong reducibility are ideal active materials for the preparation of anodes. However, the generation of Na dendrites, the adverse reactions of highly active Na and electrolyte, the volume expansion during battery discharge, the slow sodiumization/desodium kinetics problems, etc. [101], will lead to a decrease in battery life and performance changes, and even cause the security risks.

In order to solve these problems, a thin film or SEI film can be coated or formed on the surface of the Na anode to inhibit the formation of sodium dendrites. A composite anode can also be prepared to improve the sodium storage capacity. At the same time, special materials such as MXene and mesoporous carbon can be used to provide the support to reduce or eliminate the disadvantages caused by volume changes.

### 7.1. Sodium Surface Protection

In 2015, it was reported for the first time that only simple electrolyte-sodium hexafluorophosphate can achieve long-term high reversibility and non-polar electroplating/stripping of sodium anode at room temperature [102]. The formed SEI film composed of Na_2_O and NaF, the barrier effect of the SEI film on the electrolyte is conducive to the amorphous growth. Under the condition of 0.5 mA cm^−2^, 300 cycles can obtain an average coulombic efficiency of up to 99.9%. Later, it was reported that using propylene carbonate and fluoroethylene carbonate as cosolvents and high-concentration sodium salt and indium triiodide as additives can prepare a “cocktail optimization” electrolyte with high electrochemical performance and high safety for room temperature sodium–sulfur battery [54]. Through first-principles calculations and experimental characterization, it was confirmed that the fluoroethylene carbonate solvent and high salt concentration not only significantly reduced the solubility of sodium polysulfide, but also formed a strong SEI interface on the sodium anode during the cycle. Indium triiodide acts as an oxidation-reduction medium, and at the same time increases the dynamic transformation of sodium sulfide on the cathode, forming a passivation indium layer on the anode to prevent sodium sulfide corrosion.

In order to prepare a sodium anode with uniform and close interface contact that can inhibit the growth of dendrites. Zhao et al. successively proposed the anode protection method of atomic layer deposition (ALD) (Figure 7a) [103] and molecular layer deposition (MLD) (Figure 7b) coating [104]. The ultra-thin Al_2_O_3_ layer was used to protect the Na foil, which effectively inhibited the formation of mossy and dendritic Na, and significantly improved the service life of the sodium foil. Comparing the two methods of ALD Al_2_O_3_ and MLD Al_2_O_3_, it was found that the Na of MLD Al_2_O_3_ coating had a longer lifetime and the polarization curve was more stable than that of the ALD Al_2_O_3_ coating.

### 7.2. Sodium-Based Composite Anode

For problems that cannot be solved by a single sodium anode, researchers began to consider using composite anodes. Lee et al. [39] studied a sodium–sulfur battery composed of nanostructured NaS_n_-C anode, hollow carbon ball–sulfur cathode and TEGDME_4_NaCF_3_SO_3_ electrolyte, which can provide a significant capacity of 550 mAh g^−1^ and an expected theoretical energy of 550 Wh kg^−1^ density. Luo et al. [105] used capillary action to quickly melt in the carbonized wood channel to prepare a stable sodium wood composite anode. These carbonized materials have a high specific surface area, conductive and mechanically stable skeleton, which reduces the effective current density, ensures the uniformity of sodium nucleation and limits the volume change during the cycle. The results show that in the ordinary carbonate electrolyte system, the sodium wood composite anode has a lower overpotential and a stable cycle performance of 500 h at 1.0 mA cm^−2^. In contrast, the performance is better than sodium metal electrodes. Wang et al. [106] prepared a processable and moldable composite metal sodium anode made of sodium and reduced graphene oxide. Compared with sodium metal, only 4.5% reduced graphene oxide, the composite anode has better hardness, strength and corrosion resistance, and can be designed into various shapes and sizes. In ether and carbonate electrolytes, the electroplating/stripping cycle of the composite anode is significantly extended, and the formation of dendrites is suppressed. We used composite anodes in both Na-O_2_ and Na-Na_3_V_2_ (PO_4_)_3_ full batteries, and we expect that this anode can be combined with further electrolytes and anodes to develop new sodium-based batteries.

Zhang et al. [107] developed a method to simultaneously form an artificial SEI film and a metal 3D host. The conversion reaction (CR) of sodium and MoS_2_ (4Na + MoS_2_ = 2Na_2_S + Mo) at room temperature simultaneously generates an artificial SEI film and a 3D host of metal sodium. In the Na-MoS_2_ hybrid product (Na-MoS_2_ (CR)) after the conversion reaction, the generated Na_2_S is evenly distributed on the surface of the metal sodium, which can be used as an artificial SEI film to effectively prevent the growth of sodium dendrites. The residual disulfide nanometers thin slices can construct a three-dimensional host that limits metallic sodium, adapting to the volume changes of sodium to a large extent. Therefore, the Na-MoS_2_ (CR) hybrid device has a very low overpotential of 25 mV, and has very long cycle stability over 1000 cycles. This method significantly improves the stability of the sodium-based electrode.

Recently, Lou et al. [108] demonstrated a multistep templating strategy for manufacturing a layered three-layer Cu_2_S@C@MoS_2_ nanobox (Figure 7c). This synthesis method can easily adjust the structure and composition of the layered hybrid material. The mixed conformation combines the advantages of each component and facilitates the transportation and storage of electrons/Na^+^, thereby improving the sodium storage performance. This work has some enlightenment for the design of advanced anode materials for sodium ion batteries.

In addition, there are materials that can store and alloy with Na to alleviate the problem of dendrites. For example, a sodium–sulfur battery assembled with nano-Na-Sn-C as the anode, hollow carbon spherical sulfur as the cathode, and TEGDME_4_NaCF_3_SO_3_ as the electrolyte has a reversible capacity of 550 mAh g^−1^ and a theoretical energy density of 550 Wh kg^−1^, which can be recycled more than 120 h [39].

In order to reduce the generation of dendrites and unnecessary reactions of the sodium metal anode, an SEI film can be formed on the sodium surface to prevent it from being disturbed by sodium dendrites. The synthetic composite sodium anode uses its stable skeleton to slow down the volume expansion during charge and discharge. Additionally, doping with carbon and other metals to store sodium to reduce sodium dendrites.

## 8. Summary and Outlook

This article, the working principle of room temperature sodium–sulfur battery, the existing challenges and the research results of its cathode, anode, separator and electrolyte to cope with these problems are stated.

Cathode research mainly focuses on improving the conductivity of sulfur, effective sulfur fixation and sodium inhibiting dendrites. Although various carbon-based mesoporous cathode bodies can increase the sulfur utilization rate and thereby increase the initial capacity, the cycle stability is not ideal, especially for long-term cycles. Therefore, in future research, carbon-based materials integrated with metal compounds, such as MOF, metal nitrides and metal oxides, can be further studied to eliminate unnecessary capacity degradation. At the same time, the sulfur equivalent cathode material is also a good choice for sodium–sulfur batteries. In addition, an independent binder-free cathode is also a good research idea. After removing the influence of the binder on the conductivity of the cathode material, the conductivity of the cathode will be improved.

Electrolyte is a key factor affecting the performance of sodium–sulfur batteries. The performance of carbonate-based electrolytes shows a relatively high capacity and has a different voltage curve without a different platform. The combination of an ionic liquid-based electrolyte and carbonate-based electrolyte has been proved to be a better choice. Combining ionic liquids and gels can obtain electrolytes with high conductivity, independent mechanical properties and dimensional stability, which may replace carbonate electrolytes.

The membrane with ion selectivity and physical adsorption will have a good inhibitory effect on the shuttle of polysulfides. Ordinary diaphragms cannot meet these requirements. The surface of the diaphragm must be modified to allow only Na+ to pass through. The modified diaphragm has an excellent capacity retention rate. It is also possible to add an intercalation layer between the diaphragm and the cathode to reduce electrical resistance while suppressing sulfur shuttle. Due to its nanosized frame and pores, the carbon nanofoam sandwich cannot only achieve the above-mentioned purpose, but also can digest volume expansion.

Since sodium dendrites will be generated on the surface of the sodium anode, which will cause a short circuit in the battery, sodium surface protection is necessary. Depositing Al2O3 layer atomic layer or molecular layer on sodium effectively inhibits the formation of sodium dendrites and prolongs the service life of sodium. Among them, MLD Al2O3 coating has better performance. In the future, MLD can be considered as an anode coating technology. In addition, other substances can be combined with sodium to form a composite anode to improve battery performance. The Na2S produced by sodium and some metal sulfides cannot only be used as an artificial SEI membrane to inhibit the growth of sodium dendrites, it can also be used as a 3D host of sodium, digesting the volume changes of sodium and greatly improving the stability of the sodium anode. In addition, it is also an effective idea for composite anodes to manufacture layered nanoboxes and combine the advantages of various hybrid configurations to improve sodium storage performance.

## Figures and Tables

**Figure 1 molecules-26-01535-f001:**
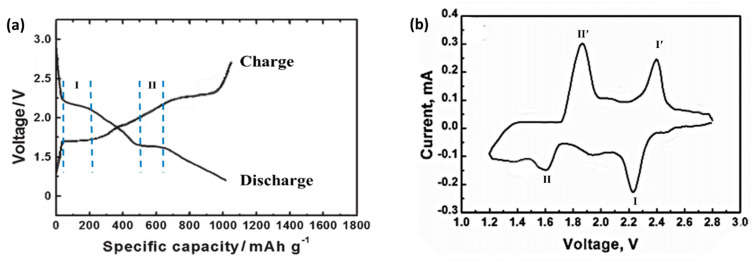
(**a**) Charge-discharge and (**b**) cyclic voltammetry (CV) curves of typical room temperature sodium sulfur batteries.

**Figure 2 molecules-26-01535-f002:**
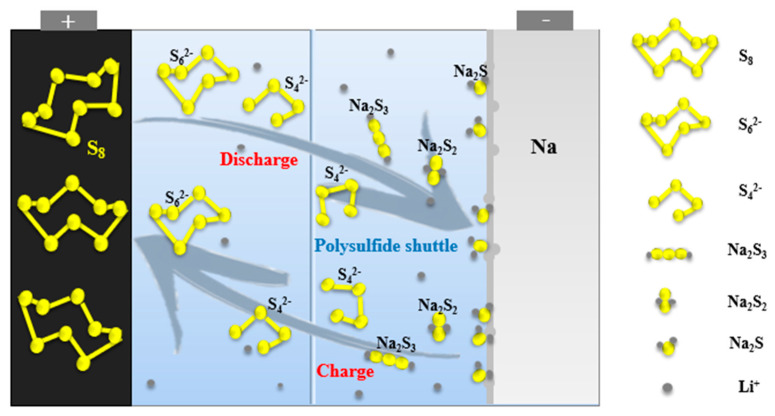
Schematic diagram of working mechanism and shuttle effect of sodium sulfur battery.

**Figure 3 molecules-26-01535-f003:**
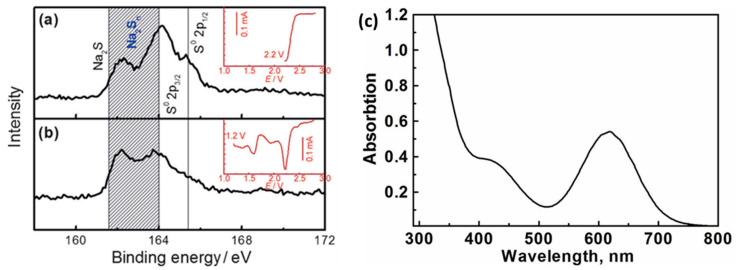
High-resolution S2p XPS spectra of the sulfur cathode being discharged to (**a**) 2.2 and (**b**) 1.2 V at a scan rate of 0.1 mVs^−1^. Insets are the voltammogram profiles of each electrode. (**c**) UV-Vis absorption spectra of the discharge product of the sulfur cathode that was discharged to 1.8 V.

**Figure 4 molecules-26-01535-f004:**
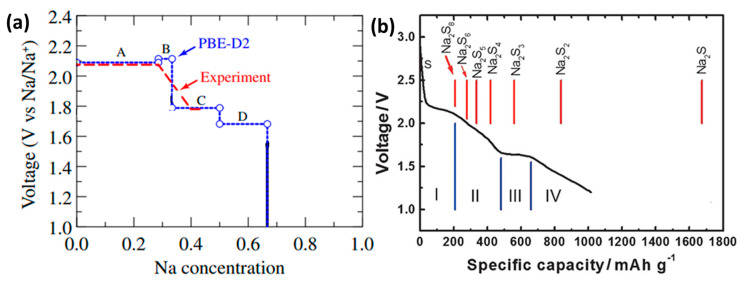
(**a**) Voltages of a Na-S battery as a function of the Na concentration in the cathode. Voltages are calculated by PBED2 ((blue) open circles with dotted line) and experimental voltages ((red) broken line) for molten Na-S batteries measured at about 280–390 °C. (**b**) Discharge curve of room temperature sodium sulfur battery.

**Figure 5 molecules-26-01535-f005:**
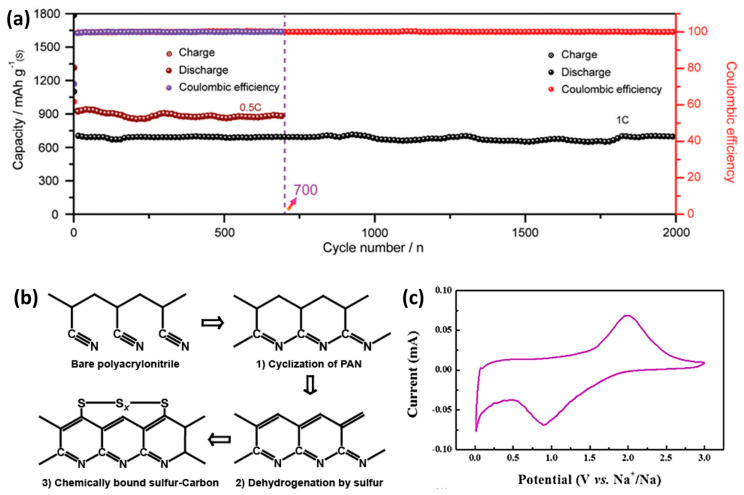
(**a**) Cycle performances of the ACC-40S electrode at 0.5 and 1 C for 700 and 2000 cycles, respectively. (**b**) Structural changes during carbonization and sulfurization of PANs. (**c**) The CV curve of the covalent sulfur-based carbonaceous (CSCM) cathode at the scan rate of 0.1 mV s^−1^ with a voltage range of 0.01–3.0 V.

**Figure 6 molecules-26-01535-f006:**
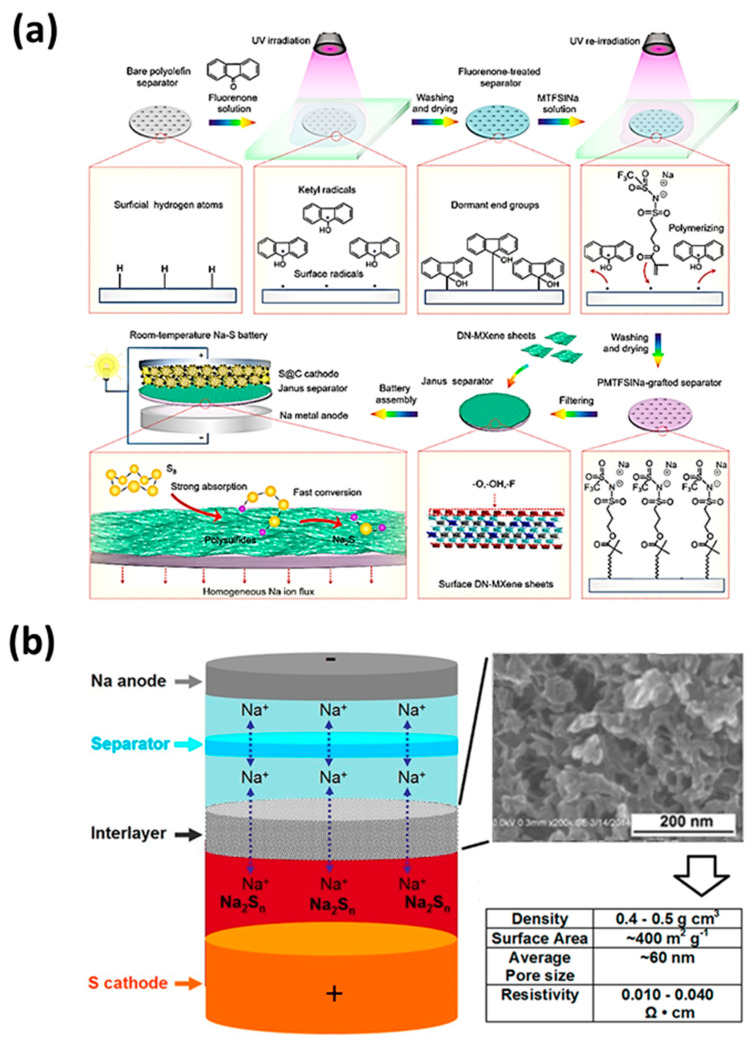
(**a**) Schematic illustration of the preparation of PMTFSINa-grafted side|DN-MXene coated side Janus separators for room temperature Na-S batteries. For the ball-and-stick model of DN-MXene, light cyan, gray, white, red and blue balls represent Ti, C, H, O and N atoms, respectively. (**b**) Schematic representation of the room-temperature sodium-sulfur battery with an interlayer, SEM image showing the structure of the carbon nanofoam and properties of the carbon nanofoam.

**Figure 7 molecules-26-01535-f007:**
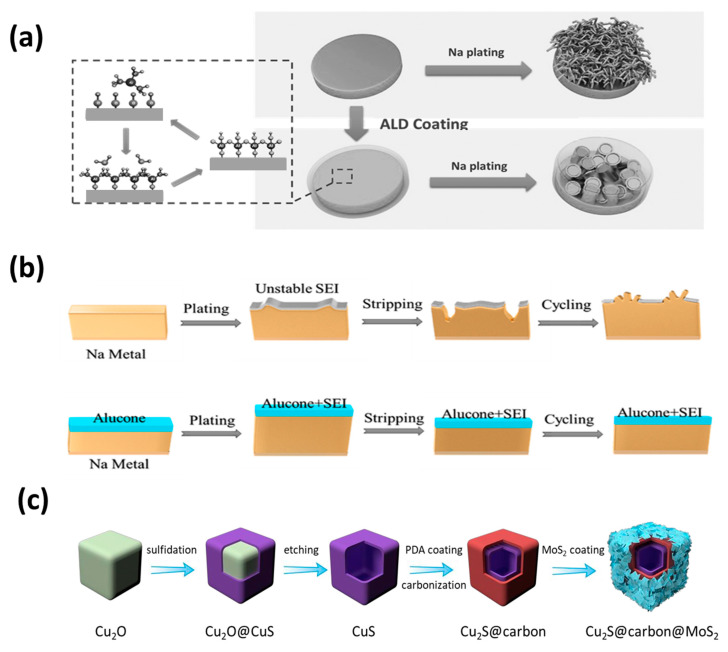
Schematic diagram of Na stripping-plating on bare Na foil and Na foil with (**a**) atomic layer deposition (ALD) coatingand (**b**) molecular layer deposition (MLD) Alucone coating (Alucone: New aluminum-based organic-inorganic composite film). (**c**) Schematic diagram of Na stripping/plating on bare Na foil and Na foil with MLD Alucone coating.

**Table 1 molecules-26-01535-t001:** The main research progress of room temperature sodium–sulfur batteries in the past six years.

Publish Date	Reference	Anode	Cathode	Electrolyte	Separator	1st dis. Capacity/mAh/g (C-Rate)	10th dis. Capacity/mAh/g (C-Rate)
2013	Lee et al. [39]	Na-Sn-C comp.	60 wt % S/Hollow C comp. (viz. 56 wt % S) 20 wt % CB 20 wt % PEO	NaCF_3_SO_3_ In TEGDME (4:1 mol %)	-	1200	600
	Hwang et al. [32]	Na	70 wt % S/C–PAN comp. (viz. 32 wt % S) 15 wt % CB 15 wt % PVDF	0.8 M NaClO_4_ in EC:DMC (1:1)	-	364	-
2013	Xin et al. [40]	Na	80 wt % S/(CNT@MPC) comp. (viz. 32 wt % S) 10 wt % CB 10 wt % PVDF	1 M NaClO_4_ in PC:EC (1:1 *v*/*v*)	-	1610	1100
2014	Bauer et al. [41]	Na	42.5 wt % S 42.5 wt % C 12 wt % PVDF 3 wt % PTFE (dry)	1 M NaClO_4_ in TEGDME	Nafion coating on PP separator	400	370
	Zheng et al. [42]	Na	80 wt % HSMC–Cu–S comp. (viz. 50 wt % S) 10 wt % CB 10 wt % CMC (in H_2_O)	1 M NaClO_4_ in EC/DMC (1:1)	-	1000	690
	Yu et al. [27]	Na	60 wt % S 30 wt % CB 10 wt % PVDF	1.5 M NaClO_4_ 0.3 M NaNO_3_ in TEGDME	-	900	600
	Yu et al. [43]	Na	MWCNT/Na_2_S_6_	1.5 M NaClO_4_ 0.3 M NaNO_3_ in TEGDME	-	945	535
	Nagata et al. [44]	Na-Sn (15:4.9 mol/mol)	50 wt % S 40 wt % SE 10 wt % AC	P_2_S_5_ (viz. 71 wt % S)	-	1456	-
				NaPS_3_ (viz.75 wt % S)		1522	
				Na_3_PS_4_ (viz.80 wt % S)		1100	
2015	Kim et al. [45]	Na	60 wt % S/C comp. (viz. 55 wt % S) 20 wt % PVDF 20 wt % Super-P	β-Al_2_O_3_ (SE)		855	674
				1M NaCF_3_SO_3_ in TEGDME (L)	PP (porous)	350	-
	Kim et al. [46]	Na	SPAN comp. (viz. 41 wt % S)	1M NaPF_6_ in EC/DEC (1:1 *v*/*v*)	GF	342	260
2016	Wei et al. [34]	Na	MCPS1 (viz. 47 wt % S)	1 M NaClO_4_ 5 v% SiO_2_–IL–ClO_4_ in EC/PC	GF	1459	762
	Fan et al. [47]	Na	70 wt % CSCM comp. (viz.18 wt % S 1 g BDTD) 20 wt % AB 10 wt % CMC in C_2_H_5_OH/H_2_O (1:2.5 *w*/*w*)	1 M NaClO_4_ in EC/DMC (6:4 *v*/*v*)	-	1000	962
	Wang et al. [48]	Na	S/iMCHS comp. (viz.46 wt % S)	1.0 M NaClO_4_ 5 wt % FEC in PC/EC (1:1 *v*/*v*)	-	1213	430
	Qiang et al. [49]	Na	N,S-HPC/S comp. (viz. 22 wt % S)	1M NaClO_4_ in EC/PC (1:1 *v*/*v*)	GF/B	400	399
	Yu et al. [50]	Na	Na_2_S/AC-CNF comp. (viz. 66 wt % Na_2_S)	1.5 M NaClO_4_ 0.2 M NaNO_3_ in TEGDME	Na-Nafion membrane	563	658
2017	Yue et al. [51]	Na-Sn-C	Na_3_PS_4_-Na_2_S-C (2:1:1 *w*/*w*) comp.	Na_3_PS(SE)	-	869	704
2018	Ye et al. [52]	Na	70 wt % N/S-OMC-5 comp. (viz.20.32 at% N 0.82 at% S) 20 wt % Super-P 10 wt % PVDF	1 M NaClO_4_ 5 wt % FEC in EC/PC (1:1 *v*/*v*)	-	1742	419
	Lee et al. [53]	Na	S/C-PAA 9:1	1M NaClO_4_ in PC	GF	623	558
				1M NaClO_4_ in PC/FEC			
	Xu et al. [54]	Na	80 wt % S@MPCF (6:4 *w*/*w*) 10 wt % Super-P 10 wt % CMCNa	2M NaTFSI in PC/FEC (1:1 *v*/*v*) with 10 mM InI_3_	GF/A	1544	1032
2019	Zhang et al. [55]	Na	70 wt % S@Fe-HC 10 wt % CB 20 wt % CMC	1M NaClO_4_ in PC/EC (1:1 *v*/*v*) with 5 wt % FEC	GF	945	630
	Li et al. [56]	Na	70 wt % Te_0.04_S_0.96_@pPAN 20 wt % SuperP 5 wt % SBR 5 wt % NaCMC	1 M NaClO_4_ in EC/DMC (1:1 *v*/*v*)	-	1816	1015
				1M NaClO_4_ in DOL/DME (1:1 *v*/*v*) with 10% FEC)		1682	868
	Zhu et al. [57]	Na	60 wt % S/CPAN comp. 40 wt % SE	PEO−NaFSI−1% TiO_2_ comp.	-	300	253
	Yan et al. [58]	Na	NiS_2_@NPCTs/S	-	-	957	508
2020	Ma et al. [35]	Na	S@Co/C/rGO	-	-	490	367
	Aslam et al. [59]	Na	S@BPCS (hollow polar bipyramid prism catalytic CoS_2_/C as a sulfur carrier)	-	GF	1347	787
	Guo et al. [60]	Na	ACC-40S (the carbon–sulfur composite electrode with 40 wt % sulfur loading)	-	GF	1492	1200
	Du et al. [16]	Na	80 wt % rGO/VO_2_/S comp. 10 wt % AB 10 wt % PVDF	1 M NaClO_4_ in TEGDME.	GF	526.2	346

TEGDME: Tetraethylene glycol dimethyl ether; PAN: Polyacrylonitrile; PEO: Polyethylene oxide; GF: Glass fiber; PVDF: Polyvinylidene fluoride; CB: Carbon black; EC: Ethylene carbonate; DMC: Dimethyl carbonate; PC: Propylene carbonate; CNT: Carbon nanotubes; MPC: Microporous carbon sheath; PTFE: Poly tetra fluoroethylene; PP: Polypropylene; HSMC: High surface area mesoporous carbon; CMC: Carboxymethyl cellulose; MWCNT: Multi-wall carbon nanotubes; SE: Solid electrolyte; AB: Acetylene Black; FEC: Fluoroethylene carbonate; HPC: Hierarchical porous carbon; AC: Activated carbon; OMC: Mesoporous carbon; PAA: Polyacrylic acid; MPCF: Multiporous carbon fibers; NaTFSI: Sodium bis(trifluoromethylthiocarbonyl)imide; SBR: Polystyrene butadiene copolymer; DOL: 1,3 dioxolane; HC: Hollow carbon; rGO: Reduced graphene oxide.

## Data Availability

Data available in a publicly accessible repository.

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
