# Peer review of "Research Progress toward Room Temperature Sodium Sulfur Batteries: A Review"

_molecules, 2021, doi:10.3390/molecules26061535_

Round 1

Reviewer 1 Report

The paper with title: “Research progress of room temperature sodium sulfur batteries: -A review “ by Yanjie Wang , Yingjie Zhang, Hongyu Cheng, Zhicong Ni, Ying Wang, Guanghui Xia, Xue Li, Xiaoyuan Zeng presents a beautiful current overview of the problems of materials as well the construction of new room temperature sodium sulphur batteries. It focuses mainly on the materials from which the individual elements of the batteries are constructed. It draws on the latest publications. Moreover, the manuscript summarized new trends, existing problems, partly it also offers the key to their solution.

The scope of review is considerable. Clarity and ordering of the issue is relatively good. The presented review provides a good overview of current knowledge about sodium sulfur batteries and I mean that also reflect the focus of the journal. The manuscript is written in clear English, the number of citations is sufficient.

But I found a very similar comparison with the same topic in the article: Recent advances in electrolytes for room-temperature sodium-sulfur batteries: A review by Mohanjeet Singh Syalia Deepak Kumarb Kuldeep Mishrac D.K. Kanchand Energy Storage Materials, Volume 31, October 2020, Pages 352-372 (https://doi.org/10.1016/j.ensm.2020.06.023) or in Revitalising sodium–sulfur batteries for non-high-temperature operation: a crucial review Yizhou Wang, Dong Zhou, Veronica Palomares, Devaraj Shanmukaraj, Bing Sun, Xiao Tang a, Chunsheng Wang, Michel Armand, Teófilo Rojo and Guoxiu Wang Energy Environ. Sci., 2020, 13, 3848-3879, (DOI: 10.1039/D0EE02203A- this article is cited as reference 13).  Unlike the sent manuscript, this both articles present a clear comparison of the properties of materials in the tables or diagrams. Such decription is very useful for the readers and is completely lack in this manuscript. If the authors would like to publish the manuscript, they should state how it differs from publications 1 and 2. The novelty should be written into the text. In addition, it is not entirely clear same perspectives from the overview, which in my opinion should also follow from the reviewed work.

Therefore I recommend the manuscript for publication after major revision mentioned below.

Recommended major revision:

  1. I recommend a comparison with the above mentioned overview papers, describe the originality of your review.
  2. I recommend including a comparison or summarization of described materials and its properties in clear tables and diagrams.
  3. After a clear summary, I recommend indicating the perspective of further research.

Recommended minor revision:

line 52: proper-ty

line 73: 489 mAhg-1 - do not divide the units

line 101: Figure 1

line 114: Is Figure 1 correct? I think Figure 2 should be mentioned.

line 132:  Figure 3

line 136: I recommend giving the number of figure - Figure 3.

line 177: morpho-logy

line 211: 5x10-30S.cm-1

line 264: Sulfide dynamics ???; e-xample

line 256: metal/carbon-sulfur carriers  - add some references

line 203: an explanation of the abbreviation SEI is missing

line 279: but the conductivity

line 287: (99%) and excellent

line 374: an explanation of the abbreviation CNTs is missing

line 427: Figure 5

line 450: Te0.04S 0.96pPAN

line 455: respectively – add references

line 490: poly-mer

line 502: interface.

line 524: Na2S

line 544: battery.;capaci-ty

line 572: poly-mer

line 603: an explanation of the abbreviation LIB is missing

line 619: poly-mer

line 664: “Through characterization and electrochemical experiments, it was proved that the formation of SEI in situ and the electrolyte under static and dynamic conditions. Stability of the anode interface.” In my opinion, is this sentence difficult to understand?

line 690: separators. This

line 758: 5 nm

line 800: hexa-fluorophosphate

line 879: stab-le

line 880: do-ping

line 887: sodium dendrites??

line 915: depo-siting

line 920: MoS2

line 1047: incomplete references 58

Reviewer 2 Report

This manuscript present a review of efforts to develop room temperature sodium sulfur batteries. A review of this nature represents a laudable undertaking. However, the manuscript will require rather massive revision to make it suitable for publication. Corrections are penciled-in directly on pages of the manuscript attached. These are illustrative of the kinds of changes needed throughout. In rewriting, careful attention should be paid to the use of articles and tenses and the mixing of intent within a sentence. Author's names and et. al. should be omitted. Lithium dendrites (Conclusions) are not formed in sodium-sulfur cells.

Reviewer 3 Report

The work “Research progress of room temperature sodium sulphur batteries: A review” by Yanjie Wang et al. summarizes the working principle and existing problems of room temperature sodium-sulphur battery. Sodium-sulphur batteries are the alternative for high cost lithium metal batteries. Also lithium metal batteries have poor safety performance, and additionally lithium resources are limited.

Authors discuss methods used to solve general scientific problems to improve the comprehensive energy storage performance of sodium-sulphur battery from four aspects of cathode, anode, electrolyte and separator.

The paper reads well and the review is clear and complete. The work is timely for the for the Molecules community.

Round 2

Reviewer 1 Report

I am very satisfied with the paper revision, practically all my recommendations were accepted by the authors.  Just pay attention to the subscripts in the  formulas of the chemical compounds in the chapter Summary and Outlook.

Reviewer 2 Report

This good topic and the manuscript is somewhat improved.